# Poly (Butylene Succinate)/Silicon Nitride Nanocomposite with Optimized Physicochemical Properties, Biocompatibility, Degradability, and Osteogenesis for Cranial Bone Repair

**DOI:** 10.3390/jfb13040231

**Published:** 2022-11-08

**Authors:** Qinghui Zhao, Shaorong Gao

**Affiliations:** Institute for Regenerative Medicine, National Stem Cell Translational Resource Center, Shanghai East Hospital, School of Life Sciences and Technology, Tongji University, Shanghai 200092, China

**Keywords:** nanocomposite, Si_3_N_4_, cellular response, degradability, bone regeneration

## Abstract

Congenital disease, tumors, infections, and trauma are the main reasons for cranial bone defects. Herein, poly (butylene succinate) (PB)/silicon nitride (Si_3_N_4_) nanocomposites (PSC) with Si3N4 content of 15 w% (PSC15) and 30 w% (PSC30) were fabricated for cranial bone repair. Compared with PB, the compressive strength, hydrophilicity, surface roughness, and protein absorption of nanocomposites were increased with the increase in Si_3_N_4_ content (from 15 w% to 30 w%). Furthermore, the cell adhesion, multiplication, and osteoblastic differentiation on PSC were significantly enhanced with the Si_3_N_4_ content increasing in vitro. PSC30 exhibited optimized physicochemical properties (compressive strength, surface roughness, hydrophilicity, and protein adsorption) and cytocompatibility. The m-CT and histological results displayed that the new bone formation for SPC30 obviously increased compared with PB, and PSC30 displayed proper degradability (75.3 w% at 12 weeks) and was gradually replaced by new bone tissue in vivo. The addition of Si_3_N_4_ into PB not only optimized the surface performances of PSC but also improved the degradability of PSC, which led to the release of Si ions and a weak alkaline environment that significantly promoted cell response and tissue regeneration. In short, the enhancements of cellular responses and bone regeneration of PSC30 were attributed to the synergism of the optimized surface performances and slow release of Si ion, and PSC30 were better than PB. Accordingly, PSC30, with good biocompatibility and degradability, displayed a promising and huge potential for cranial bone construction.

## 1. Introduction

Cranial bone constructs the neurocranium of the skull that forms a cavity and provides mechanical support to protect the brain [1]. Patients with craniofacial bone defects caused by different disorders (e.g., trauma, infection, tumor resection, and congenital malformation) suffer from problems with chewing, speech, and aesthetics [2]. Large cranial defects (e.g., critical-size defects) lead to a large area of the unprotected brain experiencing remarkable cosmetic deformity [3]. Reconstruction of the cranial defect (Cranioplasty) is commonly carried out to restore the appearance in neurosurgical surgeries, and successful reconstruction of the cranial defect is an integral step to restoring craniofacial function and improving the quality of life [4]. Cranioplasty cosmetically reshapes the cranial defect and provides a physical barrier for the protection of the cerebral structure [4]. Moreover, cranioplasty serves as a treatment measure to control the changes in the brain’s blood flow, cerebrospinal fluid, and metabolic requirements [5]. In the development of cranioplasty, some biomaterials (e.g., autograft, allograft, and synthetic biomaterial) have been applied to repair cranial defects [6]. Although autografts are still the standard for bone defect treatment, the high incidence of the donor sites mobility and the limited volume of autografts restrict the large area improvement of bone repair surgery [7]. Accordingly, current biomaterial technology develops advanced functional materials to replace autografts to construct cranial defects. Synthetic materials (metal, ceramic/cement, polymer, and composite) are used for cranial bone construction thanks to the reduced risks of resorption, infection, and reoperation compared with autografts [8].

Degradable polymers are widely applied for bone regeneration owing to good biocompatibility, degradability, mechanical properties, processability, and so on [9]. Poly(butylene succinate) (PB) is a synthetic degradable polymer that exhibits excellent biocompatibility, remarkable toughness, and non-toxicity of degradable products [10]. PB is a semi-crystalline polymer that exhibits high fracture energy and a slow degradation rate [11]. These preferable performances of PB make it a promising candidate for bone regeneration applications [10,11]. However, the major shortcoming of PB is the hydrophobic surface property because of very low surface wettability that causes poor interaction with biological fluids, which inhibits cell response [12]. Accordingly, the intrinsic hydrophobic nature and biological inertness of PB may restrict or delay cell adhesion, growth, and bone regeneration [13]. The enhancement of biological properties (e.g., wettability, degradability) of PB for regenerative medicine application is still in development. 

Human bone is a natural nanocomposite consisting of organic components (e.g., collagen) and nano-inorganic minerals (e.g., calcium phosphate) that possesses fascinating properties [14]. Inspired by the structure and composition of bone tissue, the design of nano inorganic fillers/polymer composite by integrating the advantages of both organic and inorganic phases can result in the development of high-performance nanocomposites for bone regeneration application [15]. Compared with conventional microparticles, nanoparticles with a large surface area can result in a close combination with a polymer matrix at the interface, offering enhanced mechanical performances while maintaining the favorable biocompatibility and osteoconductivity of the bioactive fillers, thereby improving protein adsorption, cells adhesion, multiplication, and osteoblastic differentiation for bone regeneration [16]. Nanocomposites of bioactive nanomaterials (e.g., bioglass, calcium phosphate, and apatite) and degradable polymers have been increasingly researched and developed for bone repair due to their superior biocompatibility, osteoconductivity, and degradability [17]. The bioactive nanocomposite is a promising class of advanced biomaterial with great potential for bone regeneration thanks to the mimic of the structure/composition and mechanical performances of natural bone tissue [18]. 

Silicon nitride (Si_3_N_4_) is a non-oxide ceramic and is regarded as a new biofunctional material with high mechanical properties, good biocompatibility, and bioactivity, which has been applied for bone repair for more than 10 years [19]. Si_3_N_4_ can be degradable in the biological environment with the slow release of silicon (Si) ions, which boosts the osteoblast response and bone regeneration [20]. In addition, the hydrophilic and negatively charged surface of Si_3_N_4_ with the bioactive groups of hydroxyl (-OH) and amino (-NH_2_) can improve the adsorption of proteins and further facilitate cell adhesion, thereby being applied as a potential biomaterial for bone regeneration application [21]. Si_3_N_4_ remarkably boosted the adhesion and multiplication of mesenchymal stem cells and improved alkaline phosphatase activity, bone-related gene expression, and bone matrix protein formation [22]. Accordingly, Si_3_N_4_ is a promising candidate for bone repair thanks to its favourable biocompatibility, osteoconductivity, hydrophilicity, and other bio-properties. 

Herein, PB/Si_3_N_4_ nanocomposites (PSC) with a Si_3_N_4_ content of 15 w% (PSC15) and 30 w% (PSC30) were fabricated through the solvent casting method, and porous PSC15 and PSC30 were prepared by solvent casting/particle leaching method. The primary goal of this paper was to produce a nanocomposite with good bioactivity and proper degradability for skull defect repair. The effects of Si_3_N_4_ content on the compressive strength, surface characteristics (e.g., topography, hydrophilicity, and protein adsorption), and degradability of PSC were investigated. The in vitro cell response (e.g., attachment and osteoblastic differentiation) to PSC was assessed, and the in vivo bone regeneration and degradability potential of porous PSC were studied using the skull defect model of rabbits. 

## 2. Materials and Methods

### 2.1. Materials and Instruments

Poly (butylene succinate) and silicon nitride particles were separately purchased from Anqing Hexing Chemical Co., Ltd., Anqing, Anhui Province, China and Shanghai Xiaohuang Nano Technology Co., Ltd., Shanghai, China. Bicinchoninic acid kit (BCA), Bovine serum albumin (BSA), Fibronectin (Fn), Fluorescein isothiocyanate (FITC), 4,6-diamidino-2-phenylindole dihydrochloride (DAPI), and ALP staining kit (BCIP/NBT) were purchased from Beyotime Biotech Co., Shanghai, China. Sodium dodecyl sulfate (SDS), simulated body fluids (SBF, pH = 7.4), and glycine were purchased from Aladdin Biochemical Technology Co., Ltd., Shanghai, China. α-MEM was purchased from Gibco, Thermo Fisher Scientific, Waltham, MA, USA. Fetal bovine serum was purchased from Hyclone, Australia. Penicillin/streptomycin (P/S), Glutaraldehyde, and Nonidet P-40 (NP-40) were purchased from Sigma, Life Technology, St. Louis, MO, USA. Cell Counting Kit-8 (CCK-8) and p-nitrophenyl phosphate (pNPP) were separately purchased from Sigma-Aldrich and Sangon, Shanghai, China. ARS solution and cetylpyridinium chloride solution were purchased from Servicebio, Wuhan, Hubei Province, China. Trizol reagent was purchased from Life Technologies, Burlington, MA, USA. The samples were characterized with scanning electron microscopy (SEM) with energy dispersive spectrometry (EDS) (S-4800, Hitachi, Tokyo, Japan), X-ray diffraction (XRD; D8, Bruker, Karlsruhe, German), and a Fourier transform infrared spectrometer (FTIR; Nicolet is50, Thermo Fisher Scientific, Waltham, MA, USA). Universal material machine (E44.304, MTS Co., Shenzhen, Guangdong Province, China). Laser confocal 3D microscope (LCM; VK-X 110, Keyence Co., Osaka, Japan) and Contact angle measurement (CAM; JC2000D1, Shanghai Zhongchen Digital Technique Apparatus Co., Shanghai, China).

### 2.2. Preparation and Characterization of Composites

The dense samples (PB, PSC15 and PSC30) were fabricated by solvent casting. In a few words, PB particles (10 g) were dissolved in Chloroform (10 mL) under stirring to prepare the PB solution. The Si_3_N_4_ powders with 0 w% (PB), 15 w% (PSC15), and 30 w% (PSC30) in the composites were then added into PB solution with continuous stirring for 6 h at room temperature for uniform dispersion. The mixture was then cast into molds (Φ6 × 6 mm for compressive strength testing and Φ12 × 2 mm for another testing) and dried in a ventilation hood for 24 h to evaporate the solvent.

The dense samples of (PB, PSC15, and PSC30) were characterized with SEM, EDS, XRD, and FTIR. The compressive strength of specimens was performed with a universal material machine. The surface roughness (Ra) and water contact angle of the samples were characterized by LCM and CAM, respectively. For the protein adsorption, the samples were placed into 24-well plates, and then the BSA (10 mg/mL) and Fn (25 μg/mL) solutions were added to the plates, respectively. After incubating at 37 °C for 5 h, the samples were extracted, and the non-absorbed proteins on the samples were removed by washing with phosphate-buffered solution (PBS, TBD, China) twice. After that, the adsorbed proteins were released by adding 1 mL SDS solution, and the protein contents were tested using the BCA assay kit.

### 2.3. Si ion Release and pH Value Variation after Samples Soaked in SBF 

The samples were immersed in simulated body fluids (SBF, pH = 7.4, Shanghai Yuanye Biotechnology Co., Ltd., Shanghai, China). At 1 d, 3 d, 7 d, 14 d, 21 d, and 28 d, the solution was collected, and the concentrations of Si ions in SBF were tested by Inductivity Coupled Plasma (ICP-OES; Agilent IC, Santa Clara, CA, USA). The release of Si ions from the specimens was also determined. Meanwhile, the pH variation of the solution during the whole period was monitored using a pH meter. 

### 2.4. Morphology, Porosity, and Water Absorption and Degradation In Vitro

The porous samples of PB, PSC15, and PSC30 were fabricated with solvent casting/particulate leaching. After different amounts of Si_3_N_4_ were uniformly dispersed into the PB solution, NaCl particles with sizes of approximately 300 μm were added into the PB solution and stirred for 10 min. Subsequently, the PB solution with NaCl was cast into the molds (Φ6 × 6 mm and Φ6 × 2 mm) and air-dried overnight. After evaporation, the samples were immersed in water for 2 days to leach NaCl particles, and the water was refreshed every 6 h. The samples were air-dried for 2 days to remove residual water. The morphology of porous samples was observed by SEM. The porosity of samples was determined with the ethanol substitution method according to the following formula: Porosity = (*V* − *V*_e_)/(*V*_0_) × 100%,
where *V*_0_ represents the total volume of samples, and *V*_e_ represents the volume of samples immersed in ethanol.

The weight of samples immersed in water for 24 h (M_w_) and the weight of dry samples (M_d_) were measured. The water absorption was obtained according to the formula:Water absorption = (M_w_ − M_d_)/M_d_ × 100%.

To assess the in vitro degradability of the porous samples, the porous samples (size of Φ12 × 2 mm) were weighed (W_d_) and then immersed into PBS solution (at 37 °C and pH 7.4) with a constant shaking speed of 60 rpm/min in an orbital shaker for various time. The samples were taken out, rinsed with water, and dried at 37 °C. Finally, the samples were weighed (W_t_). The weight loss was obtained according to the formula:Weight loss = (W_d_ − W_t_)/(W_d_) × 100%.

### 2.5. Cellular Response to Samples

#### 2.5.1. Cell Culture

The rat bone marrow mesenchymal stem (RBMS) cells were separated from the femur bone marrow of Sprague Dawley rats, the cells at passages 3–5 were cultured in α-MEM supplemented with 10% fetal bovine serum and 1% penicillin/streptomycin in a humidified atmosphere of 5% CO_2_ at 37 °C, and the medium was replaced every 2 days.

#### 2.5.2. Cell Morphology

The samples were sterilized with 75% ethanol and UV radiation and then placed in 24-well plates. The cells with a density of 5 × 10^4^ cells/well were cultured on different samples. After incubating for 1 d and 3 d, the medium was removed, and the samples were washed with PBS (3 times) and fixed with glutaraldehyde solution (0.25%) for 2 h. Then, the fixed cells were rinsed with PBS (3 times) and dehydrated by ethanol solution with various concentrations of 10 v%, 30 v%, 50 v%, 70 v%, 85 v%, 90 v%, and 100 v% for 15 min. The cell morphology was observed with SEM. Similarly, after fixation with glutaraldehyde solution (0.25%) for 2 h, the samples were gently rinsed with PBS (3 times). Subsequently, Fluorescein isothiocyanate (FITC, 400 μL) was added to stain the F-actin ring of cells for 40 min under dark conditions and rinsed with PBS 3 times. Afterwards, the nuclei of cells were stained with 4,6-diamidino-2-phenylindole dihydrochloride (DAPI, 400 μL) for 15 min and rinsed with PBS 3 times. In this way, the F-actin rings were stained green, and the nuclei were stained blue. The cell morphology was observed with confocal laser scanning microscopy (CLSM; Nikon A1R, Nikon Co., Tokyo, Japan).

#### 2.5.3. Cell Attachment and Multiplication

The attachment and multiplication of RBMS cells on different samples were investigated with a CCK-8 assay. After culturing for 6 h and 12 h, the specimens were transferred into a 24-well plate. In total, 400 μL of cell medium containing CCK-8 solution (40 μL) were added and incubated for 6 h. Subsequently, the supernatant (100 μL) was transferred into a 96-well plate, and the optical density (OD) value was measured at 450 nm with a microplate reader (MR, 384 SpectraMax, Molecular Devices, Silicon Valley, CA, USA). The OD value of the blank (without samples) was used as a control, and the cell adhesion rate was calculated according to the formula:Cell adhesion ratio = OD_s_/OD_b_ × 100%,
where OD_s_ and OD_b_ represent the OD values of cells on the samples and blank, respectively. Similarly, at 1 d, 3 d, and 7 d after culturing, the cell multiplication was determined by measuring the OD value of cells on different samples at 450 nm with MR.

#### 2.5.4. ALP/ARS Staining and Quantitative Analysis

The samples were immersed in α-MEM in a humidified atmosphere (at 37 °C) of 5 % CO_2_ for 24 h to obtain the extract. ALP/ARS staining was applied to evaluate the effects of the extract on the osteogenic differentiation of the cells. The ALP activity was evaluated by ALP staining and quantification of ALP. At 7 d and 14 d after culturing, the cells were lysed with NP-40 (1%) for 1 h and incubated with pNPP containing MgCl_2_·6H_2_O (1 mmol/L) and glycine (0.1 g/mL) for 2 h. Subsequently, the reaction was terminated by the addition of NaOH solution (0.2 mol/L). The OD value was measured at 405 nm with MR, and the total protein quantity was tested with the BCA kit. The ALP activity was calculated by dividing the measured absorbance by the total protein amount. After being cultured for 14 days, the cells were fixed with 0.25% glutaraldehyde solution for 20 min and stained with BCIP/NBT kit in the dark for 2 h. The reaction was terminated by H_2_O, and the stained samples were observed with optical microscopy. The mineralization of the extracellular matrix of cells was evaluated by ARS staining and quantification of calcium nodules. After being cultured for 14 d and 21 d, the cells were immersed in a cetylpyridinium chloride solution for 1 h to extract calcium. Subsequently, the quantitative results of calcium content were obtained by measuring the OD values for different samples at 620 nm with MR. At 21 d after culturing, the cells were fixed with 0.25% glutaraldehyde solution for 20 min and subsequently stained with ARS solution (2%) for 1 h. Then, the stained cells were washed with PBS and observed by optical microscope.

#### 2.5.5. Osteogenic-Related Gene Expressions

After the cells were cultured for 4 d, 7 d, and 14 d, the osteogenic gene expression was tested with RT-PCR. Trizol reagent was applied to extract the total RNA of the cells, and the complementary DNA (cNDA) was obtained by reversely transcribing RNA. Using the cDNA as a template, the expression of osteogenic genes (osteocalcin: OCN, alkaline phosphatase: ALP, Osteopontin: OPN, and runt-associated transcription factor 2: Runx2) was measured with the SYBR^®^ Premix Ex TaqTM system (Takara, Kyoto, Japan). Glyceraldehyde-3-phosphate dehydrogenase (GAPDH) was used as a housekeeping gene for normalization. Table 1 lists the forward and reverse primers.

### 2.6. Implantation of Samples In Vivo

#### 2.6.1. Animal Surgical Procedures

The effects of porous composites on new bone formation in vivo were determined using the rabbit skull defect model. The surgical procedures were permitted by the Animal Experiment Ethics Committee (the project identification code: TJAB03222301) of Shanghai East Hospital, School of Medicine, Tongji University. The 12 New Zealand white rabbits (around 3 kg, 8 months old) were randomly divided into 2 groups (4 w and 12 w). Pentobarbital sodium solution (3%) was used to anesthetize the rabbits by ear vein injection. The skin was sterilized with alcohol, and the cranial bone was exposed by separating the skin and cranial periosteum. Two bone defects (6 mm in diameter) were made on the bilateral sides of the rabbit skull, and PB and PSC30 were implanted into the left and right bone defects, respectively. At 4 w and 12 w after surgery, the rabbits were sacrificed with pentobarbital sodium solution (overdose) and the defective bone of the skulls was harvested and then fixed in phosphate-buffered formalin (10%).

#### 2.6.2. M-CT Images Analysis

The new bone formation for specimens was observed and imaged with microcomputed tomography (m-CT, SkyScan 1272, Bruker, Madison, WI, USA) under 80 KV with a resolution of 5 μm, and the 3D images were reconstructed. Moreover, the bone regeneration:bone volume/total volume (BV/TV), trabecular thickness (Tb.Th), trabecular number (Tb.N), and bone mineral density (BMD) were quantified by CT Analyzer (SkyScan software, CTVOX 2.1.0, Bruker, Madison, WI, USA).

#### 2.6.3. Histological Images Analysis 

After decalcifying with 10% EDTA solution for 8 w, the samples were embedded in paraffin, and histological sections with a thickness of 5 μm were obtained. Subsequently, the histological sections of H&E staining were prepared according to the standard protocol. Three microscope images were obtained with microscopy from three random areas for the sample and then evaluated with an Image-Pro Plus. The percentage of the newly formed bone area was determined by testing the number of pixels labeled through histological images. Quantitative analysis of the ratios of new bone and residual material was performed using histological images through Image-Pro Plus, Media Cybernetics, Inc., Rockville, MD, USA.

### 2.7. Statistical Analysis

Three specimens were utilized in all experiments, and the data were presented as mean ± standard deviation. Statistical significance was performed by applying one-way analysis of variance with Tukey’s Post Hoc test; *p* < 0.05 was regarded as statistically significant. The notation “*” denotes *p* < 0.05.

## 3. Results

### 3.1. Characterization of Samples

The SEM photos of dense samples are revealed in Figure 1. Under low magnification, PB showed a flat surface, while PSC15 and PSC30 exhibited rough surfaces. Under high magnification, PB also showed a flat surface, while Si_3_N_4_ particles were observed on the surface of PSC15 and PSC30. The Si_3_N_4_ particles (size of about 100 nm) were randomly distributed on PSC15 and PSC30, and the Si_3_N_4_ particles on PSC30 were more abundant than on PSC15. 

Figure 2a displays the XRD of samples. The diffraction peaks at 19.8°, 23.1°, and 29° were the peaks of PB, which were observed on both PSC15 and PSC30 [23]. No obvious peaks were observed in Si_3_N_4_, PSC15, and PSC30, indicating that Si_3_N_4_ exhibited an amorphous phase without crystalline peaks. Figure 2b illustrates the FTIR of the samples. For PB, the peak at 2980 cm^−1^ was the stretching vibration of methylene (-CH_2_-). The peak at 1718–1731 cm^−1^ was the carbonyl (-C=O), and the peak at 1363–1386 cm^−1^ was the aliphatic group (-C-O-) [24]. For Si_3_N_4_, the peaks at 3432 cm^−1^ and 1079 cm^−1^ were the amide bond (-N-H), and the peak at 973 cm^−1^ was the stretching vibration of the silicon nitrogen bond (Si-N) [25]. The peaks of PB and Si_3_N_4_ could be found in PSC15 and PSC30. Figure 2c–e revealed the EDS spectra of the samples. The C and O elements were found in PB, PSC15, and PSC30, while the Si element was seen in both PSC15 and PSC30. 

### 3.2. Physical and Chemical Properties of Samples

Figure 3a–c reveal the specimens’ compressive strength, surface roughness, and water contact angle. The compressive strength (Figure 3a) of PB, PSC15, and PSC30 was 31 ± 2.0, 43 ± 2.5, and 52 ± 3.0 MPa. The surface roughness (Figure 3b) of PB, PSC15, and PSC30 was 1.27 ± 0.10, 2.51 ± 0.10, and 3.07 ± 0.15 μm. The water contact angle (Figure 3c) of PB, PSC15, and PSC30 was 84.5 ± 5°, 72.60 ± 5°, and 59.61 ± 4°. Figure 3d reveals the protein adsorption on specimens. The BSA adsorption amount for PB, PSC15, and PSC30 was 7.63 ± 1.5%, 19.38 ± 2.0%, and 34.19 ± 2.5%. The Fn adsorption amount for PB, PSC15, and PSC30 was 5.71 ± 1.5%, 17.64 ± 2.0%, and 27.08 ± 2.5%. 

Figure 3e shows the release of Si ions from PSC15 and PSC30 into SBF after immersion for various times. The Si ions exhibited a rapid release at the early stage of immersion (within 5 d) while a slow release at the middle and late stages of immersion (from 6 d to 14 d). At 14 days, the Si ion concentrations for PSC15 and PSC30 were 0.863 mg/L and 1.572 mg/L. Figure 3f shows the pH changes after the specimens were immersed in SBF for various times. The pH values for PSC15 and PSC30 slowly increased with time. At 14 d after soaking, the pH values for PSC15 and PSC30 were 7.78 and 7.95, respectively. However, the pH values for SBF slightly decreased with time. At 14 d after, the pH value for PB was 7.13.

### 3.3. Characterization of Porous Specimens

Figure 4a–c reveal the SEM photos of the porous specimens. The macropores of all samples showed irregular morphology with pore sizes of approximately 300 μm. The porosity (Figure 4d) of PB, PSC15, and PSC30 was 63.2%, 65.9%, and 68%. Figure 4e displays the water absorption of the samples after they were immersed in water for 6 h. The water absorption for PB, PSC15, and PSC30 was 234.8%, 348.3%, and 379.3%. Figure 4f shows the weight loss of PB, PSC15, and PSC30 after being immersed in PBS for various times. At each time point, the weight loss for PSC30 was higher than PSC15, and PSC15 was higher than PB. At 84 d, the weight loss for PB, PSC15, and PSC30 was 20.58 w%, 47.63 w%, and 67.56 w%. 

### 3.4. Cell Adhesion, Multiplication, and Morphology

Figure 5 demonstrates the CLSM images of the cells on the specimens at different culturing times. The amounts of cells on PSC15 and PSC30 increased with the culturing time, while there was no obvious increase for PB. Further, the number of cells on PSC30 was higher than PSC15, and PSC15 was higher than PB. 

Figure 6a–f show the SEM photos of cells on the specimens after culturing for various times. On days 1 and 3, only a few cells were observed on PB, while some cells with filopodia spread on the surface of PSC15 and PSC30. More cells with pseudopodia spread better on PSC30 than on PSC15. The number of cells on PSC15 and PSC30 increased with time but there was only a slight increase for PB. Figure 6g reveals the cell adhesion ratio for specimens at various times. At 6 and 12 h, the cell adhesion for PSC15 and PSC30 remarkably increased with time, but there was only a slight increase for PB. Cell adhesion for PSC30 was higher than PSC15, and PSC15 was higher than PB. Figure 6h displays the optical density (OD) value (cell multiplication) of cells on specimens at 1 d, 3 d, and 7 d. The OD values for PB showed a little increase, while the OD values for PSC30 and PSC15 remarkably increased with time, indicating good cytocompatibility. At 3 and 7 days, the OD values for PSC30 were higher than PSC15, and PSC15 were higher than PB.

### 3.5. Osteoblastic Differentiation

#### 3.5.1. ALP Activity and Calcium Nodules

Figure 7a–c display the photos of the ALP staining 14 days after the cells were cultured on the samples. The intensity of ALP staining was the strongest for PSC30, followed by PSC15, and the weakest for PB. Figure 7d–f display the photos of ARS staining at 21 d after culturing. The intensity of ARS staining was the strongest for PSC30, followed by PSC15, and the weakest for PB. Figure 7g displays the quantitative results of the ALP activity of cells after culturing for various times. At 7 d and 14 d, the ALP activity for PSC30 was higher than PSC15 and PB and the lowest for PB. Figure 7 h displays the quantitative results of the calcium content of cells after culturing for various times. At 14 d and 21 d, the calcium content for PSC30 was higher than PSC15 and PB and the lowest for PB. 

#### 3.5.2. Expression of Osteoblastic Genes

Figure 8 reveals the expression of osteoblastic differentiation genes (Runx2, ALP, OCN, and OPN) of cells at various times after culturing. The osteoblastic differentiation gene expressions for PSC15 and PSC30 increased with the cultured time but showed a slight change for PB. At 7 d and 14 d, the gene expressions were the highest for PSC30, followed by PSC15, and the lowest for PB. 

### 3.6. Bone Regeneration and Material Degradation In Vivo

#### 3.6.1. M-CT Valuation

Figure 9a,b shows the macroscopic observation of the samples after being implanted into femur defects of rabbits for 4 w and 12 w. Figure 9c,d displays the m-CT reconstructed images of the samples. At 4 w after surgery, only a small amount of new bone tissue formed along the edge of PB, while some new bone tissues formed for PSC30. At 12 w after surgery, some bone tissues were seen to grow into porous PB while many new bone tissues grew into porous PSC30 and completely repaired the bone defects. 

Figure 9e–h shows the quantitative results of new bone formation (BV/TV, BMD, Tb.Th, and Tb.N) for the samples at various times. The new bone formation for both PB and PSC30 increased with time, and that for PSC30 was remarkably higher than PB at both 4 w and 12 w.

#### 3.6.2. Histological Evaluation

Figure 10a–f reveal the H&E staining images of new bone formation for the various samples at 4 w and 12 w after surgery. At 4 w, only a small amount of new bone tissues was seen in the bone defects for PB, while obvious new bone tissues were found for PSC30. At 12 w, the new bone tissues for PB slightly increased compared with 4 w, and the materials in the bone defects slightly reduced accordingly over time. Nevertheless, many new bone tissues were seen for PSC30, and the materials were reduced. Figure 10i,j shows the quantitative results of the new bone area and residual materials at 4 w and 12 w after surgery. The new bone area for PSC30 remarkably increased with time while the residual materials reduced accordingly. In addition, the new bone area for PB slowly increased with time, and the residual materials slowly reduced accordingly. The new bone area for PSC30 was remarkably higher than PB, and the residual materials for PSC30 were remarkably lower than PB.

## 4. Discussion

Interest in the application of nanocomposites with regenerative potential to repair damaged bone tissue has increased because nanocomposites containing degradable polymers and nano bioactive fillers are regarded as a mimic strategy for bone regeneration [26]. Herein, a bioactive nanocomposite was prepared for the construction of cranial bone defects by incorporating Si_3_N_4_ nanoparticles into the PB matrix. Because the Si_3_N_4_ nanoparticles reinforced PB, the compressive strength of PB, PSC15, and PSC30 gradually increased, demonstrating that Si_3_N_4_ content played a critical role in enhancing mechanical properties. Surface character is considered one of the important factors regulating cell behaviors, which exhibits significant effects on the bone–tissue response [27]. Compared to PB with flat surfaces, both PSC15 and PSC30 displayed rough surfaces thanks to the Si_3_N_4_ nanoparticles exposed on the surface. The surface roughness of PB, PSC15, and PSC30 gradually increased, demonstrating that the Si_3_N_4_ content played a critical role in the increase in surface roughness. Hydrophilicity is one of the important surface characteristics that affect cellular behaviors. However, the hydrophilic surface tends to improve cell adhesion and spreading on biomaterials compared to a hydrophobic surface [28]. Here, the hydrophilicity of PB, PSC15, and PSC30 gradually increased, revealing that the Si_3_N_4_ content played a critical role in improving hydrophilicity. Accordingly, compared with PB, the surface properties (hydrophilicity and roughness) of PSC increased, and PSC30 exhibited optimization thanks to the high content of Si_3_N_4_. 

The initial biological response to the biomaterial is protein absorption, which has been demonstrated to be a regulator between the biomaterials and cells [29]. Protein adsorption is generally affected by surface performances, especially the hydrophilicity and topography of biomaterials. Herein, compared with PB, the adsorption of proteins for PSC15 and PSC30 was remarkably enhanced thanks to the presence of Si_3_N_4_ nanoparticles. The improvement of protein adsorption for PSC15 and PSC30 was ascribed to the hydrophilicity/surface energy and topography/roughness because the hydrophilic groups (-OH, -NH_2_) of Si_3_N_4_ and rough surface of PSC with the high surface area could provide more sites for protein binding, leading to an increase in protein adsorption. Apart from the surface properties, the release of Si ions was key to stimulating cell multiplication and osteoblastic differentiation [30]; herein, the gradual release of Si ions from both PSC15 and PSC30 into PBS was ascribed to the degradation of Si_3_N_4_. The pH value for PB slightly reduced with time due to the production of acid produced by the degradation of PB, while the pH values for PSC15 and PSC30 slowly increased, causing a weak alkaline environment thanks to the production of an alkaline product by degradation of Si_3_N_4_. A weak alkaline (e.g., pH 7.4~8.0) micro-environment was demonstrated to be useful for osteoblastic differentiation and bone regeneration [31]. 

Cellular adhesion is the first response in the interaction between cells and biomaterials, which affects subsequent cell multiplication, and further affects osteoblastic differentiation and bone regeneration [32]. Herein, cell adhesion, spreading, and multiplication on PB, PSC15, and PSC30 gradually increased, indicating that the content of Si_3_N_4_ in the composites played a key role in enhancing cell adhesion and multiplication. The ALP/ARS staining and ALP activity/calcium (nodule) content of the cells cultured on the samples can be used to assess osteogenic differentiation [33]. The staining intensity of ALP/ARS for PB, PSC15, and PSC30 gradually became strong. Moreover, the ALP activity/calcium content for PB, PSC15, and PSC30 gradually increased, indicating that osteogenic differentiation improved with the increase in Si_3_N_4_ content. Further, the expression of osteogenic-associated genes of cells on the samples could be applied to evaluate osteogenic differentiation [34]. The gene expression of PB, PSC15, and PSC30 gradually increased, indicating that the osteogenic differentiation increased with the increase in the Si_3_N_4_ content. Accordingly, the content of Si_3_N_4_ in the composites played a key role in enhancing osteogenic differentiation.

Porous composites for bone regeneration should have the following characteristics: porous structures to promote cell–biomaterial interaction, cell adhesion, and growth; interconnective porous structures to boost transport of nutrients, and mass and regulated factors to allow cell multiplication, survival, and osteoblastic differentiation; pore size is essential for bone regeneration because bone growth requires an optimized pore size of approximately 300 μm [35]. Accordingly, the strategy of a combination of degradable polymer and bioactive material to create porous composites with appropriate porosity is a promising method for bone construction [35]. In this study, porous composites were prepared, and PSC30 exhibited a porous structure with a porosity of approximately 70%. Accordingly, PSC30 was used to construct cranial bone defects in rabbits. A porous composite acts as a temporary template for cell adhesion, multiplication, ensuing osteoblastic differentiation, and eventually resulting in bone regeneration [36]. Consequently, appropriate degradability of the biomaterial is an important factor that affects bone regeneration, and a degradable biomaterial can gradually disappear with time in vivo, thereby producing space for bone tissue ingrowth simultaneously [37]. Herein, the weight loss of PB, PSC15, and PSC30 in PBS increased with soaking time, indicating appropriate degradability. Moreover, the weight loss of PB, PSC15, and PSC30 increased with increasing Si_3_N_4_ content, indicating that the Si_3_N_4_ content had obvious effects on degradability. The degradation of Si_3_N_4_ particles on the macroporous walls created more micropores, which assisted the diffusion of the medium into the porous composites. This diffusion further facilitated the hydrolytic degradation of PB and the dissolution of Si_3_N_4_. Accordingly, the incorporation of Si_3_N_4_ particles into the composites increased the degradation rate of the porous composites. The goal of a degradable biomaterial is to boost tissue regeneration at the bone defect and gradually degrade in situ, eventually replacing new bone tissue [38]. The in vivo bone regenerative capability was investigated by implanting the porous composites into rabbit skull defects. The m-CT and histological results displayed that PSC30 gradually degraded and was replaced by newly formed bone tissue, while PB showed slow degradation and thus limited bone regeneration. Compared with PB, PSC30 induced rapid degradation and bone formation. Further, as shown in the histological photos, the porous composite in vivo did not lead to any adverse reactions, indicating good long-term biocompatibility. Collectively, the porous composites exhibited promising bone formation potential. 

Cell behaviors and bone tissue regeneration are closely correlative to the surface properties of biomaterials, and the capability to regulate surface characteristics (e.g., composition, topography, roughness, and hydrophilicity) can offer positive effects on responses of cells/tissues [39]. Herein, the surface properties (roughness, hydrophilicity, surface energy, and protein adsorption) of PB, PSC15, and PSC30 gradually increased with the increasing Si_3_N_4_ content, indicating that the Si_3_N_4_ content played a pivotal role in the enhancement of surface performances. Moreover, cell adhesion, multiplication, osteoblastic differentiation, and new bone formation for PB and PSC30 remarkably increased. Consequently, the significant improvement of cell response and bone regeneration for PSC30 was ascribed to the enhanced surface performance. Previous studies show that Si ions exhibit significant effects on boosting the multiplication and differentiation of osteoblasts [40]. Moreover, Si ions effectively enhance the gene expression related to the synthesis of the bone matrix, which is essential for bone tissue regeneration [41]. Accordingly, the strong osteogenic outcome induced by PSC30 was attributed to the high content of Si_3_N_4_, whose chemical composition offered the dissolution products of Si ions for bone regeneration in vivo [42]. Given the special function of Si ions, the degradation of PSC30 led to Si ion release, which resulted in the local pH in a physiological range for cell multiplication, and bone regeneration [43]. Consequently, incorporating Si_3_N_4_ into PB improved the surface properties of PSC, which remarkably stimulated the cellular responses in vitro and promoted new bone regeneration in vivo. Further, the degradability of PSC30 caused a slow release of Si ions into the local microenvironment that remarkably stimulated the responses of osteoblasts/bone tissues. The incorporation of bioactive nanoparticles into the degradable polymer created a bioactive nanocomposite that has the ability to boost cell attachment, multiplication, and new bone growth along with proper degradability. In conclusion, PSC30 with high content of Si_3_N_4_ exhibited good biocompatibility and stimulated the responses of cells/bone tissues, which were attributed to the synergism of both optimized surface properties and slow release of Si ions. PSC30 would be a promising candidate and have great potential for constructing cranial bone defects. 

## 5. Conclusions

A bioactive nanocomposite of PSC was created by the addition of Si_3_N_4_ nanoparticles into PB. In comparison with PB, the incorporation of Si_3_N_4_ significantly enhanced the compressive strength, surface hydrophilicity, roughness, and protein adsorption of PSC. Furthermore, the addition of Si_3_N_4_ accelerated the degradation of PSC, which led to the slow release of Si ions. Further, the cell response (adhesion, multiplication, and osteoblastic differentiation) to PSC was remarkably enhanced with the increase in Si_3_N_4_ content, and PSC30 displayed the highest cell response. Further, PSC30 significantly promoted bone regeneration and gradually degraded in vivo. The high content of Si_3_N_4_ in PSC led to more positive effects on in vitro cellular response and in vivo bone regeneration. Accordingly, the incorporation of Si_3_N_4_ into PB created a bioactive nanocomposite that has the ability to boost cell attachment, multiplication, and new bone growth along with proper degradability. The enhancements of cell response/bone regeneration were ascribed to the synergism of the enhanced surface performances and release of Si ions. Subsequently, PSC30 might be a promising candidate and have great potential for the construction of cranial bone defects. 

## Figures and Tables

**Figure 1 jfb-13-00231-f001:**
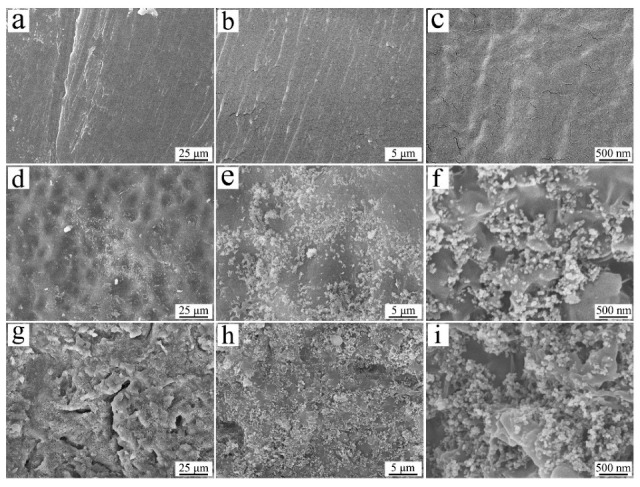
SEM photos of PB (**a**–**c**), PSC15 (**d**–**f**), and PSC30 (**g**–**i**) under various magnifications.

**Figure 2 jfb-13-00231-f002:**
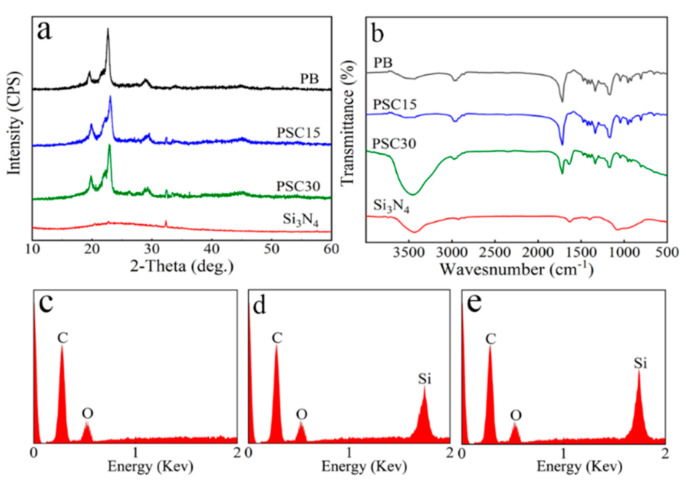
XRD (**a**) and FTIR (**b**) of the samples, and EDS of PB (**c**), PSC15 (**d**), and PSC30 (**e**).

**Figure 3 jfb-13-00231-f003:**
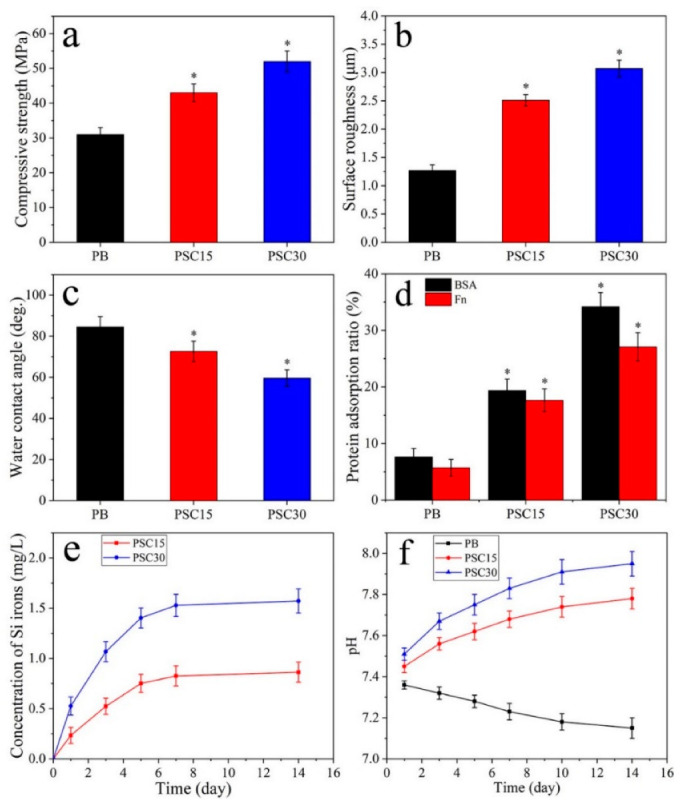
Compressive strength (**a**), surface roughness (**b**), water contact angle (**c**), and protein adsorption (**d**) of specimens, and release of Si ion (**e**) and pH change (**f**) after the samples immersed into SBF for 1 d, 3 d, 5 d, 7 d, 10 d, and 14 d (* *p* < 0.05, vs. PB).

**Figure 4 jfb-13-00231-f004:**
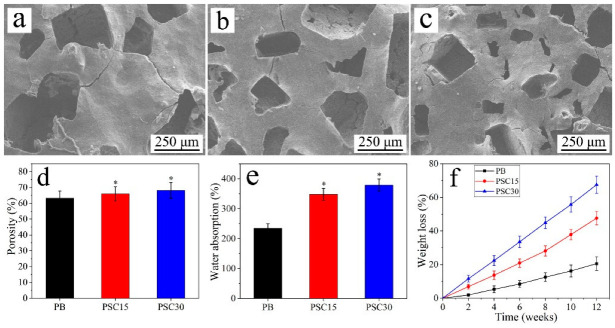
SEM photos of porous specimens of PB (**a**), PSC15 (**b**), and PSC30 (**c**), and porosity (**d**), water absorption (**e**), and weight loss (**f**) of the specimens in PBS after immersion for various times. (* *p* < 0.05, vs. PB).

**Figure 5 jfb-13-00231-f005:**
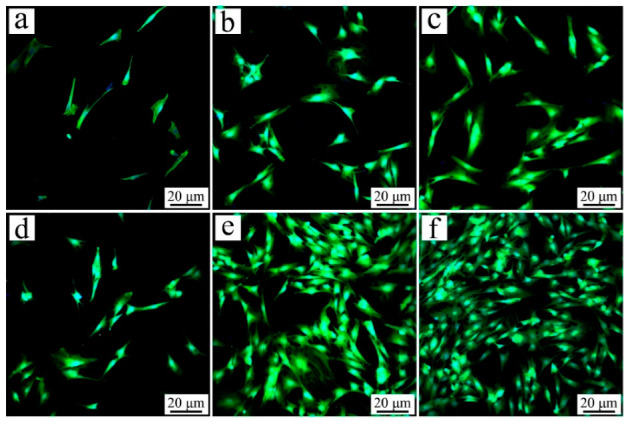
CLSM photos of RBMS cells cultured on PB (**a**,**d**), PSC15 (**b**,**e**), and PSC30 (**c**,**f**) for 1 d (**a**–**c**) and 3 d (**d**–**f**).

**Figure 6 jfb-13-00231-f006:**
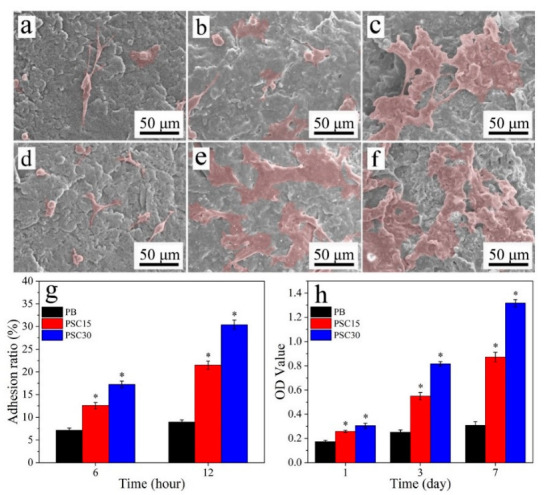
SEM photos of RBMS cells (red areas) cultured on PB (**a**,**d**), PSC15 (**b**,**e**), and PSC30 (**c**,**f**) for 1 d (**a**–**c**) and 3 d (**d**–**f**); adhesion ratio (**g**) and OD values (**h**) of cells on PB, PSC15, and PSC30 for various times (n = 3, * represents *p* < 0.05, compared with PB).

**Figure 7 jfb-13-00231-f007:**
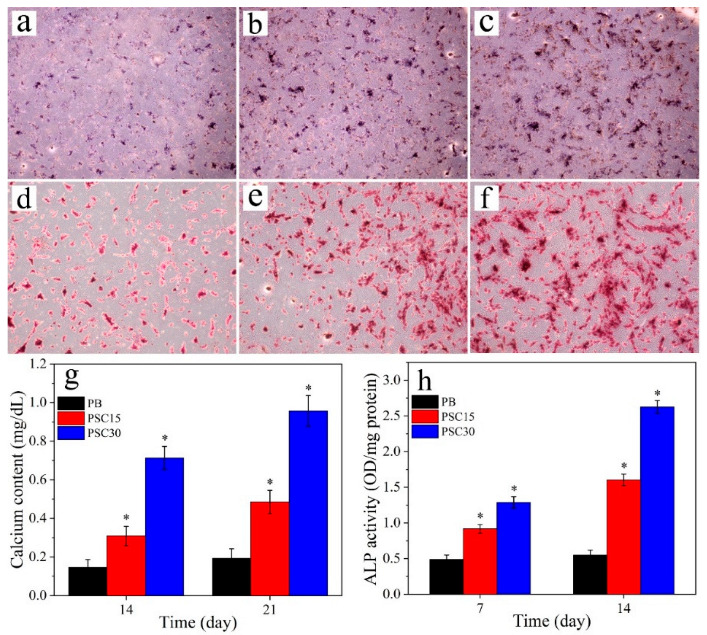
ALP staining (**a**–**c**) at 14 d after culturing and alizarin red staining (**d**–**f**) at 21 d after culturing, and quantitative results of ALP activity (**g**) and calcium content (**h**) (n = 3, * represents *p* < 0.05, compared with PB).

**Figure 8 jfb-13-00231-f008:**
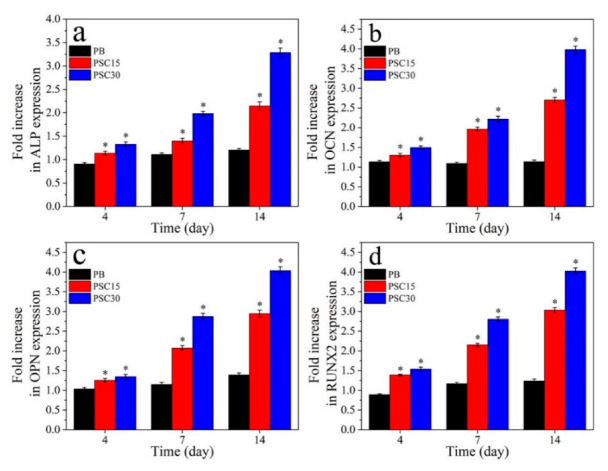
Osteoblastic differentiation genes of ALP (**a**), OCN (**b**), OPN (**c**), and Runx2 (**d**) expression of the cells cultured on PB, PSC15, and PSC30 for various times (n = 3, * represents *p* < 0.05, compared with PB).

**Figure 9 jfb-13-00231-f009:**
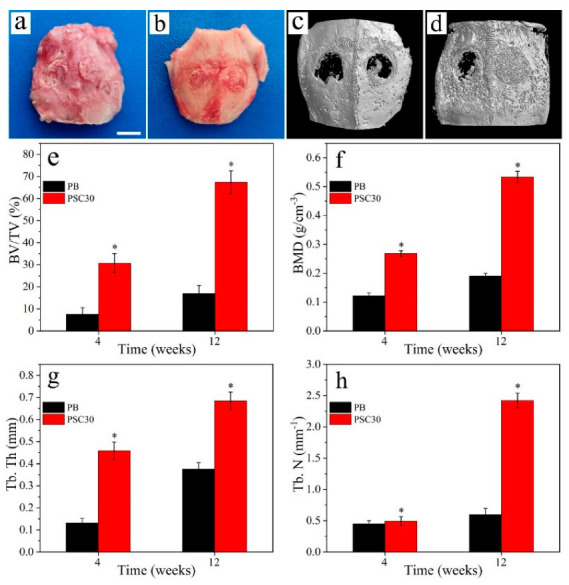
Macroscopic observation ((**a**,**b**), scale = 10 mm) and m-CT images (**c**,**d**) of PB and PSC30 after implanted into bone defects of rabbit femur for 4 w and 12 w, and quantitative results of BV/TV (**e**), BMD (**f**), Tb.Th (**g**), and Tb.N (**h**) (n = 3, * represents *p* < 0.05, compared with PB).

**Figure 10 jfb-13-00231-f010:**
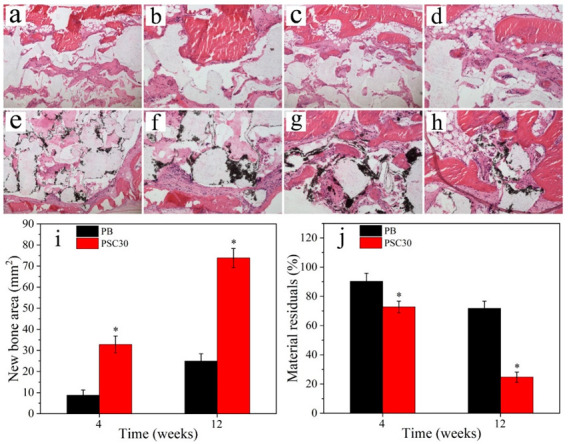
Histological images for PB (**a**–**d**) and PSC30 (**e**–**h**) after surgery for 4 w (**a**,**b**,**e**,**f**) and 12 w (**c**,**d**,**g**,**h**); quantitative results of the new bone area (**i**) and residual materials (**j**) (n = 3, * represents *p* < 0.05, compared with PB).

**Table 1 jfb-13-00231-t001:** Primer sequences.

Gene	Primers Sequence (F Was Forward, R Was Reverse)
GAPDH	F: CCTGCACCACCAACTGCTTA R: GGCCATCCACAGTCTTCTGAG
ALP	F: GGATCAAAGCAGCATCTTACCAG R: GCTTTCCCATCTTCCGACACT
OPN	F: GTCCCTTGCCCTGACTACTCT R: GACATCTTTTGCAAACCGTGT
OCN	F: CAGACAAGTCCCACACAGCA R: CCAGCAGAGTGAGCAGAGAG
Runx2	F: ATCCAGCCACCTTCACTTACACC R: GGGACCATTGGGAACTGATAGG

## Data Availability

The data presented in this study are available on request from the corresponding author.

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
