# Peer review of "Poly (Butylene Succinate)/Silicon Nitride Nanocomposite with Optimized Physicochemical Properties, Biocompatibility, Degradability, and Osteogenesis for Cranial Bone Repair"

_jfb, 2022, doi:10.3390/jfb13040231_

Round 1

Reviewer 1 Report

This is an intersting work. The authors should revise their paper based on the below comments:

Abstract

the Si3N4 content increasing: Describe the concentration

PSC with 30 w% Si3N4: Introduce the range of concentration before.

optimized physicochemical properties: list out

 cytocompatibility: Briefly explain how?

new bone formation for porous PSC obviously increased: Provide evidence how do you claim this?

proper degradability: time of degradation?

 but also improved the degradation: Howlong?

 the enhancements of cellular responses and bone regeneration of PSC30: Bit more details here, provide the clear statement before howmany concentration used and PSC30 was better than XXX?

1.Introduction: This section is clear and good.

2. Materials and methods

2.1. Materials and instruments: From" Poly (butylene succinate) particles from Anqing Hexing Chemical to CT Analyzer (SkyScan software, CTVOX 2.1.0, Bruker, USA). Microscopy (TE2000U, Nikon, Japan)".: None of the sentences here is complete. Make a complete sentence.

into PB solution with continuously stirring: How long? Temp?

After immersed into simulated body fluids : Protocol for SBF?

The rat bone marrow mesenchymal stem (RBMS) cells were separated from rat femur: Purity of the cells? Pheotypes marker using negative control, How many passages used for study?

After incubating for 1 d and 3 d: Why specific 3 days?

The ALP/ARS staining was applied to evaluate osteogenic differentiation of the cells in the extract. At 7 d and 14 d after culturing: It is not clear how long the ALP and ARS staining done? Based on the method, both staining for 7 and 14 days?

At 14 d and 21 d after culturing: So ARS staining done for 14 and 21 d? but previously stated 7 and 12 d? The authors should repeat ALP and ARS staining in the same day to keep consistent and comparable results.

2.5. Morphology, porosity and water absorption and degradation in vitro : Move this section before invitro cell culture work.

Figure 4. Describe the stain details

Figure 4: Provide FITC and DAPI and merged images here as described in method (Ref:162: stained with FITC (40 min) for and DAPI (15 min)).

Figure 5. SEM photos: Wounder why the cells look pink color in SEM image.

Figure 6.: Why the ALP and Alizarine red stain done in different days? It is advisable to do these experiments in the same way.

Figure 6.: ALP staining (a,b,c) at 14 d after culturing: In the method, it was stated "At 7 d and 14 d after culturing", It is important to provide the ALP images for 7 and 14 days as described in method

Figure 6.: alizarin red staining (d,e,f) at 21 d after culturing: Based on method "At 14 d and 21 d", provide images for 14 and 21 days. However, it is advisable, keep consistent experiment setup for ALP and alizarin red stain. 

Figure 6.:quantitative results of ALP activity (g) and calcium content (h).: Again inconsistency, 14 and 21 d for Calcium and 7 and 14 d for ALP, why? Keep same set-up.

 calcium content (h). : As per image, its not 'hour' here, its day.

ALP activity (g): What is unit g ?

Figure 6.: Provide statistical details in legends.

Figure 7. Osteoblastic differentiation genes of ALP (a), OCN (b): In Fig.6, the calcium content was measured for 14 and 21 days. So its better to provide the OCN data for same days for comparison. 

Figure 9. Macroscopic observation (a,b) and m-CT images (c,d) after PB and PSC30 implanted: Why didn't the author used PSC15 for invivo? in this case, what was the reason to use PSC 15 in in vitro study?

Author Response

Comments and Suggestions for Authors

This is an interesting work. The authors should revise their paper based on the below comments:

Abstract:

the Si3N4 content increasing: Describe the concentration

Responses: The Si3N4 content increasing (0 w%, 15 w% and 30 w%).

PSC with 30 w% Si3N4: Introduce the range of concentration before.

Responses: The Si3N4 content in the composite ranged from 15 w% to 30 w%.

optimized physicochemical properties: list out

Responses: We listed out optimized physicochemical properties (compressive strength, surface roughness, hydrophilicity and protein adsorption).

cytocompatibility: Briefly explain how?

Responses: Already described cytocompatibility in the abstract (the cell adhesion, multiplication and osteoblastic differentiation on PSC were significantly enhanced with the Si3N4 content increasing in vitro).

new bone formation for porous PSC obviously increased: Provide evidence how do you claim this?

Responses: We have revised these

The m-CT and histological results displayed that the new bone formation for porous PSC30 obviously increased compared with PB.

Proper degradability: time of degradation?

Responses:  

The m-CT and histological results displayed that the new bone formation for PSC30 obviously increased compared with PB, and PSC30 displayed proper degradability (75.3 w% at 12 weeks) and gradually replaced by new bone tissue in vivo.

But also improved the degradation: How long?

Responses: Fig.7f showed the weight loss of PB, PSC15 and PSC30 after immersed into PBS for various time. At 84 d, the weight loss (the vitro degradation) for PB, PSC15 and PSC30 were 20.58 w%, 47.63 w% and 67.56 w%.

Moreover, and PSC30 displayed proper degradability (75.3 w% at 12 weeks) and gradually replaced by new bone tissue in vivo, while PB displayed low degradability (28.2 w% at 12 weeks).

The enhancements of cellular responses and bone regeneration of PSC30: Bit more details here, provide the clear statement before how many concentration used and PSC30 was better than XXX?

Responses: Yes, the enhancements of cellular responses and bone regeneration of PSC30 were attributed to the synergism of the optimized surface performances and slow release of Si ion, and PSC30 were better than PB.

  1. Introduction: This section is clear and good.

Responses: Yes.

  1. Materials and methods

2.1. Materials and instruments: From" Poly (butylene succinate) particles from Anqing Hexing Chemical to CT Analyzer (SkyScan software, CTVOX 2.1.0, Bruker, USA). Microscopy (TE2000U, Nikon, Japan)".: None of the sentences here is complete. Make a complete sentence.

Responses: Yes, we have revised this section according to the suggestions from the reviewer.

into PB solution with continuously stirring: How long? Temp?

Responses: PB solution with continuously stirring for 6 hours at room temperature.

After immersed into simulated body fluids: Protocol for SBF?

Responses: Simulated body fluid (SBF) was purchase from Shanghai Yuanye Biotechnology Co., Ltd., China.

The rat bone marrow mesenchymal stem (RBMS) cells were separated from rat femur: Purity of the cells? Pheotypes marker using negative control, How many passages used for study?

Responses: The RBMS used in this study were isolated from Sprague Dawley rats and cultured according to the previous literature [1-5], and the purity of third-passage cells obtained by this method could reach up to 99% (99.42% positive-antigen CD44 and 0.02% negative-antigen CD45) [1]. So, we did not characterize the cell purity and cells at passage 3-5 were used for subsequent experiments. We have added relevant information in the revised manuscript.

[1] Hu, B.T.; Chen, W.Z. Eur. Rev. Med. Pharmaco. 2018, 22, 7156-7163.

[2] Ruckh, T.T.; Carroll, D.A.; et al. J. Funct. Biomater. 2012, 3, 776-798.

[3] Liu, A.Q.; Lin D.; et al.  Biomaterials. 2021, 272, 120718.

[4] Zhong, G.; Yao, J.; Huang, X.; et al. Bioact. Mater. 2020, 5, 871-879.

[5] Liu, X.; Bao, C.Y.; et al. Acta Biomater. 2016, 42, 378-388.

After incubating for 1 d and 3 d: Why specific 3 days?

Responses: Yes, the cell morphology was observed with CLSM after incubating for 1 d and 3 d according to the references [6-7]. At 3 days after incubating, the cells showed good multiplication if the materials possess good cytocompatibility.

We did these experiments according to some references are listed as following:

[6] Luo, T.; Liu, J.; et al. Int Endod J. 2018, 51, 779-788.

[7] Wang, F.; Wang, et al. J Mater Sci Technol. 2022, 133, 195-208.

The ALP/ARS staining was applied to evaluate osteogenic differentiation of the cells in the extract. At 7 d and 14 d after culturing: It is not clear how long the ALP and ARS staining done? Based on the method, both staining for 7 and 14 days?

Responses: The ALP is a marker at the early stage (e.g., from 7 to 14 days) of osteogenic differentiation, while the calcium nodules (mineralization) is an evidence at the late stage (from 14 to 21 days) of osteogenic differentiation.

Therefore, the ALP staining was only done at 14 d after culturing, while the ARS staining was only done at 21 d after culturing.

We did these experiments according to the refences [8-9].

We have rewritten this section and made it clear.

Refences

[8] Hou, F.S.; Jiang, W.; et al. Chem Eng J. 2021, 427, 132000.

[9] Ino, K.; Onodera, T.; et al.Electrochim Acta. 2018, 268, 554-561.

At 14 d and 21 d after culturing: So ARS staining done for 14 and 21 d? but previously stated 7 and 14 d? The authors should repeat ALP and ARS staining in the same day to keep consistent and comparable results.

Responses: The ALP is a marker at the early stage (from 7 to 14 days) of osteogenic differentiation, while the ARS staining of calcium nodules (mineralization) is an evidence at the late stage (from 14 to 21 days) of osteogenic differentiation.

Therefore, the ALP staining was only done at 14 d after culturing, while the ARS staining was only done at 21 d after culturing.

We did these experiments according to the refences [10-11].

We have rewritten this section and made it clear.

[10] Qiao, Y.S.; Liu, X.Z.; et al. Adv Healthc Mater. 2019, 9, 1901239.

[11] Sattary, M.; Rafienia, M.; et al. Mat Sci Eng C-Mater. 2019, 97, 141-155.

2.5. Morphology, porosity and water absorption and degradation in vitro: Move this section before invitro cell culture work.

Responses: Yes, we have moved this section before in vitro cell culture work according to the suggestions from the reviewer.

Figure 4. Describe the stain details

Responses: We have provided the stain details in revised manuscript. Briefly, after fixed with glutaraldehyde solution (0.25%) for 2 h, the cells were rinsed gently with PBS for 3 times. After that, Fluorescein isothiocyanate (FITC, 400 μL) was added to stain the F-actin ring of the cells for 30 mins under dark conditions, following rinsed with PBS for 3 times. Then, the nuclei of the cells were stained with 4,6-diamidino-2-phenylindole dihydrochloride (DAPI, 400 μL) for 8 min and rinsed with PBS for 3 times. In this way, the nuclei of the cells were stained into blue and the F-actin ring of the cells were stained into green.

Figure 4: Provide FITC and DAPI and merged images here as described in method (Ref:162: stained with FITC (40 min) for and DAPI (15 min).

Responses: We have provided the merged images of FITC and DAPI images here as described in method, we think it is OK.

We think it not necessary to provide the FITC and DAPI images, respectively (some many images in one paper is not proper).

Figure 5. SEM photos: Wounder why the cells look pink color in SEM image.

Responses: Yes, the cells in SEM image were treated by software (photoshop) in order to clearly see the cells on the samples. The original images obtained with SEM have no color, and the morphology of the spreading cells and surrounding materials are difficult to distinguish under this magnification and resolution. The color was post-treated with image-processing software (photoshop) to make the cells more visible.

Figure 6.: Why the ALP and Alizarine red stain done in different days? It is advisable to do these experiments in the same way.

Responses:  The ALP is a marker at the early stage (from 7 to 14 days) of osteogenic differentiation, while the ARS staining of calcium nodules (mineralization) is an evidence at the late stage (from 14 to 21 days) of osteogenic differentiation.

Therefore, the ALP staining was only done at 14 d after culturing, while the ARS staining was only done at 21 d after culturing.

We did these experiments according to the refences [10-11].

[10] Qiao, Y.S.; Liu, X.Z.; et al. Adv Healthc Mater. 2019, 9, 1901239.

[11] Sattary, M.; Rafienia, M.; et al. Mat Sci Eng C-Mater. 2019, 97, 141-155.

Figure 6.: ALP staining (a,b,c) at 14 d after culturing: In the method, it was stated "At 7 d and 14 d after culturing", It is important to provide the ALP images for 7 and 14 days as described in method

Responses: The ALP staining was only done at 14 d after culturing (we only provided representative images for 14 days).

Moreover, the ALP activity was tested at 7 d and 14 d after culturing.

We did these experiments according to some references are listed as following:

[8] Hou, F.S.; Jiang, W.; et al. Chem Eng J. 2021, 427, 132000.

[9] Ino, K.; Onodera, T.; et al.Electrochim Acta. 2018, 268, 554-561.

Figure 6.: alizarin red staining (d,e,f) at 21 d after culturing: Based on method "At 14 d and 21 d", provide images for 14 and 21 days. However, it is advisable, keep consistent experiment setup for ALP and alizarin red stain.

Responses: The ARS staining was only done at 21 d after culturing (we only provided representative images for 21 days).

Moreover, the content of calcium (calcium nodules) was tested at 14 d and 21 d after culturing.

We did these experiments according to the refences [10-11].

[10] Qiao, Y.S.; Liu, X.Z.; et al. Adv Healthc Mater. 2019, 9, 1901239.

[11] Sattary, M.; Rafienia, M.; et al. Mat Sci Eng C-Mater. 2019, 97, 141-155.

Figure 6.:quantitative results of ALP activity (g) and calcium content (h).: Again inconsistency, 14 and 21 d for Calcium and 7 and 14 d for ALP, why? Keep same set-up.

Responses: The ALP is a marker at the early stage (from 7 to 14 days) of osteoblastic differentiation, while the calcium nodules (mineralization) is an evidence at the late stage (from 14 to 21 days) of osteoblastic differentiation.

Therefore, the ALP activity was tested at different time (7 d and 14 d) after culturing, while the content of calcium (calcium nodules) was tested at different time (14 d and 21 d) after culturing.

We did these experiments according to some references are listed as following [12-13]:

[12] Sun, Y.M.; Hu, C.; et al.Mat Sci Eng C-Mater. 2019, 103, 109836.

[13] Ding, X.X.; Li, X.; et al.ACS Biomater Sci Eng. 2019, 5, 4574-4586.

Calcium content (h): As per image, its not 'hour' here, its day.

Responses: It is a mistake, we have corrected. In Calcium content (h), “h” is the sequence Number of Fig.6, it is not unit.

ALP activity (g): What is unit g ?

Responses: It is a mistake, we have corrected. In ALP activity (g), “g” is the sequence Number of Fig.6, it is not unit.

Figure 6.: Provide statistical details in legends.

Responses: Yes, we have provided statistical details in legends in the revised manuscript.

Figure 7. Osteoblastic differentiation genes of ALP (a), OCN (b): In Fig.6, the calcium content was measured for 14 and 21 days. So its better to provide the OCN data for same days for comparison.

Responses: The expression of osteoblastic differentiation genes of ALP, OCN, OPN and Runx2 take occurs at the early stage (from 4 to 14 days) of cell culturing.

Therefore, the expression of osteoblastic differentiation genes of ALP, OCN, OPN and Runx2 was tested at 4, 7 and 14 days.

We did these experiments according to some references are listed as following [14-15]:

[14] Wang, C.; Xu, D.L.; et al. Mat Sci Eng C-Mater. 2021, 131, 112531.

[15] Zhang, Z.C.; Jia, B.; et al. Bioact Mater. 2021, 6, 3999-4013.

Figure 9. Macroscopic observation (a,b) and m-CT images (c,d) after PB and PSC30 implanted: Why didn't the author used PSC15 for in vivo? in this case, what was the reason to use PSC 15 in in vitro study?

Responses: The primary goal of this paper was to produce a nanocomposite with good bioactivity and proper degradability for skull defect repair.

The in vitro experiments demonstrated that the physical and chemical properties (e.g., compressive strength, topography, hydrophilicity and protein adsorption) of PSC30 was better than PSC15.

Moreover, the in vitro cell experiments confirmed that the cytocompatibility (e.g., attachment and osteoblastic differentiation) of PSC30 was better than PSC15. Therefore, we only chose PSC30 for in vivo experiments compared with PB (PB as a control).

It is not necessary to use PSC15 for in vivo experiments.

Reviewer 2 Report

The manuscript entitled: Poly (butylene succinate)/silicon nitride nanocomposite with
optimized physicochemical properties, biocompatibility, degradability and osteogenesis for cranial bone repair’ by Qinghui Zhao et al. provides evidence for poly (butylene succinate) (PB)/silicon nitride (Si3N4) nanocomposites (PSC) as a new scaffold material with adequate physico-chemical- and osteo-inductive properties, biocompatibility and degradability to support cranial bone repair in a small animal model.

Major comments:

11.)    For the reviewer it seems unclear why BS/Si3N4 nanocomposite (PSC) with Si3N4 content of 15% (PSC15) and 30% (PSC30) were fabricated and what is the motivation to modify into porous trait by solvent casting leaching method? Please explain at the end of the discussion. Otherwise, the reader has the assumption that just another scaffold is presented.

22.)    There is no information given on the mesenchymal stem cells (MSCs) used from bone marrow. Which passage and mesenchymal stem cell specific markers were used?

33.)    Also, no information on animal ethics is provided and which strain of rats were used?

44.)    The assay of osteogenic differentiation ALP/ARS is very simple, the expression of osteogenic-related genes is also minimalistic and GAPDH is not the right house-keeping gene in MSC-derived cells since MSCs react to cultivation in high glucose with changes in GAPDH expression. This will obscure your results.

55.)    Rat MSC-derived cells within different scaffolds were implanted into New Zealand white rabbits to cover 6 mm skull bone defects. Why was this xeno-genic model used?

66.)    It would be of interest whether the newly formed harbors neo-vessel formation and how developed these vessels were.  Is there a lumen formed and are the newly formed vessels connected to the rabbit’s circulation?

77.)    The SEM images of Figure 1 in my PDF version is of very poor quality.

88.)    The same is true with Figure 4. In the legend to Figure 4 the green staining has no prevalence.

The cells should also be stained with Phalloidin to visualize stress fiber formation. This would be a readout for cell response to the surrounding scaffold. This is a mandatory experiment.

99.)    Figure 5 gives selected images of MSC-derived cells on different scaffolds but there is no information how many experiments were done and what is mend by adhesion ratio or OD values. These are measures with no particular value.  Images a-f seem to be technically stained. No information is given to the red color applied to the images.

110.)  Figure 6 – ALP staining is also of poor quality in my copy.

111.)  Figure 6h ALP activity is not an adequate measure.

112.)  Figure 7 uses the wrong house keeping gene to normalize. GAPDH is suboptimal in MSCs. Please offer an alternative – ideally PPIA as specific house-keeping gene.

Li X, Yang Q, Bai J, Yang Y, Zhong L and Wang Y: Identification of optimal reference genes for quantitative PCR studies on human mesenchymal stem cells. Mol Med Rep 11: 1304-1311, 2015

113.)  The HE images of Figure 10 are of poor quality. Furthermore, in total experiments in 6 rabbits were performed with the two timepoints 4/12 weeks – three rabbits per timepoint were included is that correct? To clarify this issue, information should be provided.

Minor comments:

11.)    The results are structured explicitly to the Figures provided and not to the content found. Headlines should be provided with the major findings as content.

22.)    The language has lots of obscure descriptions not understandable for the interested reader.

Author Response

Comments and Suggestions for Authors

The manuscript entitled: Poly (butylene succinate)/silicon nitride nanocomposite with

optimized physicochemical properties, biocompatibility, degradability and osteogenesis for cranial bone repair’ by Qinghui Zhao et al. provides evidence for poly (butylene succinate) (PB)/silicon nitride (Si3N4) nanocomposites (PSC) as a new scaffold material with adequate physico-chemical- and osteo-inductive properties, biocompatibility and degradability to support cranial bone repair in a small animal model.

Major comments:

11.) For the reviewer it seems unclear why BS/Si3N4 nanocomposite (PSC) with Si3N4 content of 15% (PSC15) and 30% (PSC30) were fabricated and what is the motivation to modify into porous trait by solvent casting leaching method? Please explain at the end of the discussion. Otherwise, the reader has the assumption that just another scaffold is presented.

Responses: Herein, PBS/Si3N4 nanocomposite (PSC) with Si3N4 content of 15 w% (PSC15) and 30 w% (PSC30) were fabricated through solvent casting method, and porous PSC15 and PSC30 were prepared by solvent casting/particle leaching method.

We chose addition of Si3N4 content of 15 w% (PSC15) and 30 w% (PSC30) into PB according to our pre-experiments because Si3N4 content was more than 30 w%, the mechanical properties of the nanocomposite (PSC) is very low. Therefore, we chose Si3N4 content of 30 w% as maximum.

The primary goal of this paper was to produce a nanocomposite with good bioactivity and proper degradability for skull defect repair.

The effects of Si3N4 content on the compressive strength, surface characters (e.g. topography, hydrophilicity and protein adsorption), and degradability of PSC were investigated. The in vitro cell response (e.g., attachment and osteoblastic differentiation) to PSC were assessed, and the vivo bone regeneration and degradability potential of porous PSC were studied by using the skull defect model of rabbits.

In in vitro cells experiments, the dense sample (PB, PSC15 and PSC30) were used to study the effects of Si3N4 content on cell response.

In in vivo implantation experiments, the porous samples (PB and PSC30) were used to study the effects of addition of Si3N4 on bone regeneration and bone ingrowth into the porous samples and degradability of the porous samples in vivo as compared with PB.

The dense samples were used to study the physicochemical properties and cytocompatibility of samples in vitro, while the porous samples were used to study the degradability and new bone ingrowth into the porous samples in vivo

22.) There is no information given on the mesenchymal stem cells (MSCs) used from bone marrow. Which passage and mesenchymal stem cell specific markers were used?

Responses: The MSCs used in this study were isolated from Sprague Dawley rats and cultured according to the previous literature [1-5]. We did not characterize the mesenchymal stem cells with specific markers. The third-passage cells obtained by this method showed positive-antigen CD44 (99.42%) and negative-antigen CD45 (0.02%) [1]. We have supplemented relevant information in revision manuscript.

[1] Hu, B.T.; Chen, W.Z.  Eur. Rev. Med. Pharmaco. 2018, 22, 7156-7163.

[2] Ruckh, T.T.; Carroll, D.A. J. Funct. Biomater. 2012, 3, 776-798.

[3] Liu, A.Q.; Lin D. Biomaterials. 2021, 272, 120718.

[4] Zhong, G.; Yao, J. Bioact. Mater. 2020, 5, 871-879.

[5] Liu, X.; Bao, C.Y. Acta Biomater. 2016, 42, 378-388.

33.) Also, no information on animal ethics is provided and which strain of rats were used?

Responses: The surgical procedures were permitted by the Animal Ethics Committee from School of Life Sciences and Technology of Tongji University.

The 12 New Zealand white rabbits (around 3 kg, 8 months old) were in animal experiments.

The rat bone marrow mesenchymal stem (BMS) cells were separated from rat femur bone marrow (The male Sprague Dawley rats).

44.) The assay of osteogenic differentiation ALP/ARS is very simple, the expression of osteogenic-related genes is also minimalistic and GAPDH is not the right house-keeping gene in MSC-derived cells since MSCs react to cultivation in high glucose with changes in GAPDH expression. This will obscure your results.

Responses:  The ALP/AR staining and ALP activity/calcium (nodule) content of the cells cultured on the samples are powerful methods to assess the osteogenic differentiation in vitro [16]. ALP is a marker of early osteogenic differentiation while calcium nodules occur at late stage of osteogenic differentiation [17]. Therefore, in this study, the osteogenic differentiations of cells cultured on different materials were determined at different time.

Furthermore, the expression of osteogenic-related genes and in vivo experiments were employed to confirm the osteogenic ability of different materials. Thanks for your valuable advice, and we will take more methods into consideration to make the exposition stronger and more complete in our subsequent experiments.

The GAPDH is an enzyme distributed in cells of various tissues and is expressed at high levels in almost all tissue cells. Since its expression in the same cell or tissue is generally constant and rarely influenced by external inducers, GAPDH is widely used as house-keeping gene of BMSC [18-20].

Thanks for your constructive advice, and we will take the use of other more appropriate genes as house-keeping gene into consideration in our subsequent experiments.

[16] Wu, S.H.; Xiao, Z.L.; et al. Int. J. Mol. Sci. 2018, 19, 2171.

[17] Boanini, E.; Pagani, S.; et al. J. Funct. Biomater. 2022, 13, 65.

[18] Xie, J.N.; Cheng, S.; et al. J. Funct. Biomater. 2022, 13, 50.

[19] Yan, Y.F.; Chen, H.; et al. Biomaterials. 2019, 190, 97-110.

[20] Wang, F.; Wang, M.Y.; et al. J. Mater. Sci. Technol. 2022, 133, 195-208.

55.) Rat MSC-derived cells within different scaffolds were implanted into New Zealand white rabbits to cover 6 mm skull bone defects. Why was this xeno-genic model used?

Responses: We did not used the Rat MSC-derived cells within different scaffolds, which were implanted into New Zealand white rabbits.

Only the scaffolds (without cells) were implanted into New Zealand white rabbits.

The effects of porous composites on new bone formation in vivo were determined by utilizing the rabbit skull defect model.

The objective of this experiment was to study the bone regeneration (bone tissues ingrowth into porous scaffolds) and degradability potential of porous PSC30 vivo.

66.) It would be of interest whether the newly formed harbors neo-vessel formation and how developed these vessels were. Is there a lumen formed and are the newly formed vessels connected to the rabbit’s circulation?

Responses: Yes, it would be of interest of your suggestions. Unfortunately, we did not do these tests. It is very interesting and significant, we will do these in the next step.

77.) The SEM images of Figure 1 in my PDF version is of very poor quality.

Responses: Yes, we have improved the quality of SEM images of Figure 1.

88.) The same is true with Figure 4. In the legend to Figure 4 the green staining has no prevalence.

Responses: Yes, we have improved the quality of Figure 4. Moreover, we have adjusted color of green staining.

The cells should also be stained with Phalloidin to visualize stress fiber formation. This would be a readout for cell response to the surrounding scaffold. This is a mandatory experiment.

Responses: It is very interesting and good suggestion. However, we are sorry that we did not do this experiment. We will try to do this experiment in the next step.

99.) Figure 5 gives selected images of MSC-derived cells on different scaffolds but there is no information how many experiments were done and what is mend by adhesion ratio or OD values. These are measures with no particular value. Images a-f seem to be technically stained. No information is given to the red color applied to the images.

Responses: Yes, we have provided the information how many experiments (3 times) were done.

Herein, cell adhesion and multiplication on different samples were evaluated according to CCK-8 assay. The cell adhesion (figure 5g) was calculated with OD value of cells after culturing for 6 and 12 h. The OD value of cells in blank (without samples) was used as controls, and cell adhesion rate was calculated according to the formula: Cell adhesion ratio = ODs/ODb× 100%. Where ODs and ODb represent the OD values of cells on the samples and blank, respectively.

Similarly, at 1 d, 3 d and 7 d after culturing, the cell multiplication (figure 5h) was determined by measuring the OD value of cells on different samples. In general, higher adhesion ratio or OD value indicates greater cell adhesion or multiplication ability.

We have provided no particular value in the revised manuscript. In all experiments, three specimens were utilized for each analysis and each sample was analyzed in triplicate.

Yes, the Images a-f (the cells in SEM image) were technically stained by software (photoshop) in order to clearly see the cells on the samples. The SEM images in figure 5 were obtained from randomly selected areas in different samples and three random regions on each specimen were observed. The color was post-added with image-processing software to make the cells more visible. We have added explanation in the diagram of figure 5. (The red areas indicate the cells spread on different samples)

110.) Figure 6 – ALP staining is also of poor quality in my copy.

Responses: Yes, we have improved the quality of ALP staining.

111.)  Figure 6h ALP activity is not an adequate measure.

Responses: The ALP activity was measured at 7 d and 14 d after culturing.

We did these experiments according to some references are listed as following [21-22]:

[21] Nunez, C.C.; Gaete, D.A.; et al. Materials. 2021, 14, 2684.

[22] Cui, X.; Huang, C.C.; et al. Bioact Mater. 2021, 6, 3801-3811.

112.) Figure 7 uses the wrong house keeping gene to normalize. GAPDH is suboptimal in MSCs. Please offer an alternative – ideally PPIA as specific house-keeping gene. Li X, Yang Q, Bai J, Yang Y, Zhong L and Wang Y: Identification of optimal reference genes for quantitative PCR studies on human mesenchymal stem cells. Mol Med Rep 11: 1304-1311, 2015

Responses: The GAPDH is an enzyme distributed in cells of various tissues and is expressed at high levels in almost all tissue cells. Since its expression in the same cell or tissue is generally constant and rarely influenced by external inducers, GAPDH is widely used as house-keeping gene of BMSC [23-27].

Thanks for your constructive advice, and we will take the use of PPIA as house-keeping gene into consideration in our subsequent experiments.

[23] Boanini, E.; Pagani, S.; et al. J. Funct. Biomater. 2022, 13, 65.

[24] Xie, J.N.; Cheng, S.; et al. J. Funct. Biomater. 2022, 13, 50.

[25] Yan, Y.F.; Chen, H.; et al. Biomaterials. 2019, 190, 97-110.

[26] Wang, F.; Wang, M.Y.; et al. J. Mater. Sci. Technol. 2022, 133, 195-208.

[27] Sun, Y.H.; Tan, J.; et al.ACS Biomater Sci Eng. 2018, 4, 2552-2562.

113.)  The HE images of Figure 10 are of poor quality. Furthermore, in total experiments in 6 rabbits were performed with the two timepoints 4/12 weeks – three rabbits per timepoint were included is that correct? To clarify this issue, information should be provided.

Responses: We have improved the quality of the HE images of Figure 10.  

In animal model experiments, the 12 New Zealand white rabbits were performed with the two timepoints 4/12 weeks. It is a mistake, we have corrected these in the revised manuscript.

Minor comments:

11.) The results are structured explicitly to the Figures provided and not to the content found. Headlines should be provided with the major findings as content.

Responses: Yes, we have provided some Headlines with the major findings as content.

22.) The language has lots of obscure descriptions not understandable for the interested reader.

Responses: Yes, we have carefully checked and revised the manuscript according to the reviewers.

Round 2

Reviewer 1 Report

No further comments